# Influence of Durum Wheat Bran Particle Size on Phytochemical Content and on Leavened Bread Baking Quality

**DOI:** 10.3390/foods10030489

**Published:** 2021-02-24

**Authors:** Nabeel T. Alzuwaid, Denise Pleming, Christopher M. Fellows, Barbara Laddomada, Mike Sissons

**Affiliations:** 1School of Science and Technology, University of New England, Armidale, NSW 2351, Australia or nab1972eel@yahoo.com (N.T.A.); or cfellows65536@gmail.com (C.M.F.); 2NSW Department of Primary Industries, Tamworth Agricultural Institute, 4 Marsden Park Road, Tamworth, NSW 2340, Australia; 3Department of Biology, University of Dhi-Qar, Nasiriyah, Iraq; 4NSW Department of Primary Industries, Wagga Wagga Agricultural Institute, Pine Gully Road, Wagga, NSW 2650, Australia; denise.pleming@dpi.nsw.gov.au; 5Istituto di Scienze delle Produzioni Alimentari (I.S.P.A.), C.N.R., Via Monteroni, 73100 Lecce, Italy; barbara.laddomada@ispa.cnr.it

**Keywords:** durum wheat, antioxidants, phenolic acids, phytosterols, bread, bran

## Abstract

Wheat bran is a conventional by-product of the wheat milling industry mainly used for animal feed. It is a rich and inexpensive source of phytonutrients, so is in demand for fibre-rich food products but creates quality issues when incorporated into bread. The purpose of this study was to characterize the physicochemical properties and phytochemical composition of different size durum bran fractions and show how they impact bread quality. Durum wheat (*Triticum durum* Desf.) was milled to create a coarse bran fraction (CB), which was further ground into a finer fraction (FB) which was sieved using four screens with apertures 425, 315, 250, 180, and <180 µm to create a particle size range of 1497 to 115 µm. All fractions contained phytosterol with highest in the 180 and FB, while total phenolic acids and antioxidant capacity was highest in CB and 425. Use of the fractions in a leavened common wheat (*T. aestivum* L.) bread formula at 10% incorporation negatively impacted bread loaf volume, colour, and texture compared to standard loaves, with CB having the least impact. Results suggest that to combine the highest phytochemical content with minimal impact on bread quality, bran particle size should be considered, with CB being the best choice.

## 1. Introduction

Wheat bran is the major by-product of wheat milling, with an estimated annual production of 150 million metric tonnes mostly used in the feed industry [1]. Wheat bran is inexpensive and a rich source of vitamins, dietary fibre, phytosterols, proteins, phytochemicals, β-glucan, lignans, arabinoxylans, and residual starch, with many of these components having reported health benefits [1]. Studies have been conducted on incorporating wheat bran in cereal-based foods such as breads, cake, noodles, and pasta [2]. Whole grain products are reported to have health benefits such as reducing the risks of cardiovascular disease, colorectal cancer, and type 2 diabetes mellitus because the outer layers (bran and germ) of whole grains are rich in fibre and bioactive compounds [3]. Increased consumer demand for higher fibre has led to a demand for fibre-rich food products. However, inclusion of wheat bran in cereal-based foods can produce undesirable impacts on sensory and textural properties, including detrimental effects on dough rheology, reduced loaf volume, impaired bread crumb texture, and imparting dark colour and undesirable flavor to foods [4].

Several techniques have been used in an attempt to overcome these drawbacks, such as enzymatic modification, and physical treatments including superfine grinding, autoclaving, microfluidisation, soaking, and hydrothermal treatment [4,5]. Reducing the particle size of wheat bran by grinding and subsequent separation by sieving have been used widely by researchers to study the effects of adding wheat bran fractions on bread quality [4,6,7]. According to Kim et al. [8], reduction of wheat bran particle size has positive effects such as increasing cellulose and hemicellulose digestibility, decreasing insoluble and increasing soluble dietary fibre and thus improving bran functionality. Reduction of bran particle size also increases extractability of phytochemicals from bran [9]. Pasqualone et al. [10] reported that moderate addition of micronised bran (126 to 300 µm) to durum bread has a less negative impact on specific volume, crumb hardness, resilience, and chewiness compared to coarse bran. The same trend was confirmed by Wang et al. [6], who reported that Chinese bread supplemented with fine wheat bran (~100 µm) enhanced the quality of bread in terms of a larger specific volume and good crumb structure. However, some studies revealed that coarse and medium size bran (400–800 µm) had less effect on bread properties than fine bran, which reduced specific volume and darkened crumb colour of bread [4,7]. Variation in these studies may be attributed to several factors: differences in the chemical composition of wheat bran used, the method used to prepare wheat bran fractions, and variation in the procedure used for bread baking [4,11]. While information about the best bran size range to use in foods to maximise bread quality and taste is controversial, information about optimizing bread quality together with phytochemical contents, such as antioxidants, phenolic acids, and phytosterol known for their health effects, is not well documented. We chose durum wheat bran due to the paucity of information about impacts of bran from durum on leavened bread quality.

The objective of this study was to characterise the physicochemical properties and phytochemical composition of durum bran fractions and show how particle size impacts leavened bread technological properties by their inclusion in a common wheat bread mix.

## 2. Materials and Methods

### 2.1. Materials

Bran fractions were obtained from durum wheat described in detail previously [12] and summarised in Figure 1. All fractions are referred to as CB, FB, 425, 315, 250, 180, and <180 throughout the manuscript. Fractions were stored in sealed plastic bags at −20 °C until use and allowed to reach room temperature and mixed before analysis.

### 2.2. Proximate Composition

Proximate composition included moisture, protein, ash, and starch content of bran fractions determined as described previously [13].

### 2.3. Bran Particle Size Distribution

Particle size distribution of bran fractions were analysed using a Mastersizer Hydro 3000LV laser diffraction analyser as described previously [12]. Median particle size diameter, D50 (value of particle diameter at 50% in cumulative distribution) for fraction is the mean of three data values. Results are volume percent versus particle size (µm).

### 2.4. Phytochemical Analysis

The phytosterol content and composition of the bran fractions were analysed in duplicate according to Alzuwaid et al. [13].

Total phenolic acids (comprising soluble and insoluble fraction, TPA) were extracted from bran fractions and analysed by HPLC according to details described previously [12]. All analyses were performed on duplicate extracts. Methanol soluble antioxidants were assayed using the radical DPPH (2,2-diphenyl-1-picrylhydrazyl) scavenging capacity as described previously [14]. The results were expressed in terms of Trolox equivalent antioxidant capacity (TE), as mmol Trolox equivalents per kg dry weight (mmol TE/kg). Samples were extracted in duplicate, and each extract was assayed in duplicate, and means are presented.

### 2.5. Preparation of Bread Mixtures and Analysis of Bread Quality

To determine the effect of adding wheat bran fractions on bread properties, bran fractions obtained from durum wheat were combined with a commercial baking flour (*T. aestivum*), (BF; Perfection baking flour (specifications: 12% protein, 1.2%, fat, 0.65% ash), Allied Mills, Rhodes, Australia) at 10% blends *w*/*w* (14% mb) in individual containers and stirred thoroughly before addition of further ingredients. Loaves were baked according to the standard Australian rapid dough method [15] with some modifications. Two 180 g doughs were produced from a 250 g flour formulation including (on flour weight basis): 1.4% Lesaffre instant yeast (*Saccharomyces cerevisiae*), 2% salt (NaCl, LR grade), 1% Pure Vita canola oil (sourced from local supermarket) and 0.5% bread improver (including calcium sulphate and ammonium chloride as yeast foods, ascorbic acid as oxidiser to improve volume and crumb structure, xylanase to improve starch and gluten hydration, and alpha-amylase to provide sustainable source of sugars to yeast). Bake water absorption for each bran fraction blend was calculated from farinograph water absorption adjusted by a factor of minus 3 to produce optimum dough consistency in the bakery formulation. Doughs were mixed to approximately 1 min beyond optimum development in a DoughLAB 2500 (Perten Instruments, Macquarie Park, Australia) fitted with 300 g temperature-controlled bowl, then rested 5 min before scaling to two 180 g pieces and passing through a Mini Moulder (MONO Equipment, Swansea, UK) at 5 mm roll gap and 35 mm pressure board settings. Moulded doughs were placed in sealed containers to ferment at 30 °C for 10 min, then passed through the moulder again, placed in open bake tins and proofed at 34 °C and 85% RH (relative humidity) for 70 min before baking at 215 °C for 20 min in a Rotel II bakery oven (Moffat, Melbourne, Australia). Loaf height was measured before loading into oven (proof height) and at removal (oven height), with % oven spring calculated as oven height minus proof height divided by oven height ×100. After cooling for 45 min, loaves were weighed, and loaf volume determined using a pup volumeter (National Manufacturing, Lincoln, NE, USA). The following day external loaf appearance was assessed subjectively before removing a 20 mm slice from centre of loaf for instrumental texture profile analysis using TAXT2 texture analyser (Stable Micro Systems, Godalming, UK), fitted with a 35 mm flat cylindrical probe and test speed setting of 1 mm/s, hold time 3 s, and compression distance of 40%. Parameters measured were firmness (g), chewiness (undefined), resilience (%), and cohesiveness (%) according to Texture Technologies Corp and Stable Micro Systems standard texture profile analysis method [16]. Moisture was determined using two-stage bread moisture method AACC Approved Method 44-15.02 [17]. Crumb colour was measured on the central portion of each face of the sliced loaf using a Minolta CR-410 (Konica Minolta, Osaka, Japan) fitted with a 50 mm head, and the result of duplicate measurements averaged. Baking was performed in triplicate on a single day, providing six loaves of each sample for full analysis including comparison 100% wheat flour (BF) loaves.

### 2.6. Statistical Analyses

Statistical Analysis System, (GenStat 11.1, VSN International Ltd.) was used for analysis of variance. Means were compared to test for significant differences (*p* < 0.05 and 0.001) using the least significant difference statistic (LSD).

## 3. Results

### 3.1. Bran Particle Size Distribution

The particle size distribution of the bran fractions is shown in Figure 2. CB had the highest median diameter of 1497 µm and a broad range while <180 had the lowest median size, 115 µm, with the other fractions having D50 FB (350 µm), 425 (641 µm), 315 (481 µm), 250 (357 µm), 180 (247 µm). There was overlap between FB and most of the fractions. The sieved FB fractions show distributions that overlap with increasing median particle size going from 115 to 641 µm for fractions <180 to 425.

### 3.2. Proximate Composition

The chemical composition of bran fractions is shown in Table 1. All fractions had similar moisture content except CB, which was about 2–3% higher. Protein content of FB fractionated by sieving increased with reduction in particle size with highest content found in <180 (19.2%). Ash content showed little variation between bran fractions. Total starch of bran fractions tended to increase with reduction in particle size from 315 to <180.

### 3.3. Antioxidant Capacity

Antioxidant capacity (AO) of methanolic extracts from bran fractions are presented in Figure 3. All bran fractions contained AO capacity, with highest levels found in CB followed by 425. Of the remaining fractions, no significant differences were found between 315, 250, and 180 while FB and <180 were slightly higher.

### 3.4. Phenolic Acids

In this study, six phenolic acids were investigated, *p*-hydroxybenzoic, syringic, vanillic, *p*-coumaric, sinapic, and ferulic acid (Figure 4). The fraction with the highest TPA was 425 and the least <180 with significant differences between other fractions. The main phenolic acid was ferulic acid, representing over 80% of TPA.

### 3.5. Phytosterol Content

Analysis of phytosterols in the bran fractions are shown in Table 2. β-sitosterol is the main component in all fractions while stigmasterol was in the lowest amount. The highest levels of all components were found in the 180 and FB fractions, while there were few differences in all phytosterol components between CB, 425, 315, 250, and <180 fractions.

### 3.6. Effects of Bran Level and Particle Size on Bread-Making Properties

A 100% BF loaf was compared to a flour mix containing 10% *w/w* of each fraction (Table 3 and Table 4). To clearly delineate the effect of bran addition on bread quality, high quality bakers flour (BF) was used in this study. A 10% level of inclusion was chosen to optimally discriminate between samples-the effects of lower rates are more easily masked by the base flour and higher rates produce uniformly poor loaves. Loaf volume, specific volume and oven spring significantly (*p* < 0.001) deteriorated with bran fraction addition (Table 3). The LV and oven spring closest to control values was found with CB and the fraction having the greatest negative impact was <180. The FB, 315, 250 fractions had equivalent LV while 180 was lower. Although there was a high correlation between median particle size and LV (r = 0.72, *p* < 0.05), the 425 fraction did not follow the trend toward increased LV with a larger median particle size with low LV equivalent to <180.

Bran incorporation made the crumb colour less bright (lower *L**) and more reddish and yellow (greater *a** and *b**) (Table 4). The most deleterious impact on colour was using fraction 425 which produced a duller and more reddish colour compared to the other fractions. Bread with fine fractions (180 and <180) produced loaves closest in brightness (*L**) to the control, while CB inclusion produced the least yellow (*b**) crumb of the fractions.

Bread texture was affected by addition of the different bran fractions with significantly increased bread firmness and chewiness, especially with the finer fractions (Table 4). Resilience and cohesiveness decreased with all bran fractions, again with the biggest impact from the finer fractions.

Both external and cross sections of loaves are shown in Figure 5. Subjective analysis by a trained technician revealed the control loaf to have very good external character, soft, resilient crumb and evenly distributed cell structure with fine cell walls resulting in an overall rating of very good. Adding CB did not seriously detract from external character or crumb texture but did increase cell wall thickness and unevenness in cell distribution gaining an overall rating of satisfactory. FB impacted on external character, and with firm crumb, thick cell walls and uneven distribution was rated unsatisfactory along with 425, 315 and 250. The 180 loaf is poor, exhibiting a slightly rough exterior with very poor oven spring, firm crumb, thick walled and unevenly distributed cell structure. The worst loaf is made with <180 bran fraction being rated very poor, with a slightly rough exterior and very poor oven spring, very firm crumb, and thick walled, unevenly distributed cell structure.

## 4. Discussion

### 4.1. Composition of Bran Fractions

Moisture content of CB was higher than other fractions, which were all similar, due to moisture losses incurred in subsequent grinding/sieving processes. The coarse bran obtained by Bühler milling contains some endosperm that cannot be removed by milling operation and remains attached to the bran particles. With particle size reduction and sieving, endosperm particles become detached and this increases the protein and starch contents in the finer fractions (Table 1) also reported by Curti et al. [18]. Ash content of bran fractions was in the range of 3–4 g/100 g with no consistent trend and although one might expect CB and 425 to have higher ash due to the higher ratio of bran to endosperm than finer material, this was not observed. It is possible that total dietary fibre content of the fractions would differ but this was not measured.

Wheat bran is a rich source of antioxidants and increasing their content in foods is desirable as their consumption has been associated with a reduced risk of developing cardiovascular disease and cancer [2]. The DPPH assay measures single electron transfer to determine the antioxidant reducing capacity and the best electron transfer is correlated with less processing or structural alteration of the food [9]. CB and 425 exhibited the highest AO activity (Figure 3) consistent with them having less endosperm (Table 1). These data are in good agreement with the findings by Brewer et al. [9] who reported that coarse and medium bran fractions exhibited higher AO than fine wheat bran. The total phenolic acid content tended to be higher in CB and larger bran fractions than 180 and <180 fractions (Figure 4) following a similar trend to AO.

Phytosterols are one of the principal components of plant cell walls which have health benefits for humans, such as reducing the risk of elevated cholesterol [19]. Campesterol, β -sitosterol, sitostanol, and campestanol are considered the most common forms of phytosterols in wheat. While the finer bran fractions had the least TPA and AO, the FB and 180 fractions had higher total phytosterol levels than all the other bran fractions. Reducing bran particle size will change its functional properties such as increased water holding capacity, phenolic acids and antioxidant activity, because of a change in structure and surface characteristics [20]. This might explain the higher concentration of phytosterol in FB and 180 as size reduction could increase the extraction of phytosterols, however the <180 fraction exhibited the least phytosterol content. Wheat bran is composed of various structurally and compositionally different tissues, and the final character of individual fractions will be influenced by the milling process used to achieve those fractions [21]. A sieving process, as used in this study, may produce less uniformity of chemical composition across fractions than grinding methods where the sample is wholly recovered for each particle size fraction produced [7]. It is therefore plausible that interactions exist between both inherent phytosterol levels in the fractions and extraction efficiency due to particle size.

### 4.2. Impact of Bran Fractions on Bread Quality

There are three factors to consider when adding the different bran fractions created to a bread formulae and their impact on baking quality (i) role of particle size of the fraction (ii) the composition of the different fractions and (iii) the quantity of fraction incorporated. Bran incorporation was limited in this study to 10%, a level chosen to optimally discriminate between the effects of different bran particle sizes. Typically, bran binds more water compared to the starch and protein flour components [22], due to the presence of numerous hydroxyl groups in the bran interacting with water through hydrogen bonds [2]. There is no clear evidence that addition of any of the bran fractions at 10% level interfered with the gluten matrix enough to change the farinograph dough development time and stability (data not shown), although finer fractions (D50 < 400µm) had shorter bakery mix times (225–237 s) compared to coarser fractions, especially CB (255–285 s). There are multiple studies evaluating the impact of fibres on dough stability with conflicting results reviewed by Ktenioudaki and Gallagher [23]. As previously discussed, the finer fractions have relatively higher concentrations of starch and protein compared to coarser fractions. This is an indication that increased fibre content in the CB competes more with the wheat proteins for available water, thus slowing gluten hydration of these flour-bran mixtures.

Bran (un-fractionated) addition to bread is known to generally reduce loaf volume, producing a denser, less aerated structure contributing to a harder and darker crumb [8,24,25] and this was apparent in this study using 10% *w/w* replacement in the bread mix (Table 3 and Table 4). Several reasons have been proposed for negative effects of fibre (i) dilution of the gluten forming proteins [23] (ii) restriction of the available water for gluten development due to the higher water binding capacity of bran resulting in negative impacts on gluten water absorption [25] and (iii) disruption of the gluten network development [26]. These factors lead to gluten becoming less extensible, producing a gluten network with lower ability to hold and stabilise gas cells and retain the loaf structure during fermentation and baking. In addition, during baking, starch gelatinisation starts sooner in bread fortified with bran than regular bread, resulting in an earlier setting of the bread crumb structure and thus a smaller final loaf volume. As wheat bran is rich in phenolic acids compared to flour, another factor that may affect gluten negatively is formation of covalent linkages between cysteine and phenolic acids, especially ferulic acid [11,27].

Our results show that reducing the bran particle size leads to a reduction in LV and specific volume with significant correlations between median particle size and LV, specific volume and oven spring (r = 0.71, 0.74, 0.81, *p* < 0.05, respectively). This is thought to be related to the negative effects of increased bran particle surface area per unit volume leading to more particle/gluten interactions. Noort et al. [11] hypothesised that bran can interfere with gluten development reducing the gluten proteins available for gas cell stabilisation while Wang et al. [28] suggested that bran affects the properties of the gluten which becomes stiffer and less extensible affecting gas retention and reducing loaf volume. Our results indicate that coarse bran had the least effect on bread quality, while the fraction <180 had the biggest negative impact, consistent with the results of Noort et al. [11] Although not measured, if we assume coarse bran fractions contain more fibre than finer bran, this would be expected to increase water absorption of CB-flour doughs, and this was observed in our study with fractions of D50 < 400 µm having farinograph water absorption of 1.0 to 1.4% less than coarser fractions. Increased fibre may also be expected to cause more deterioration of bread quality, but we obtained the best results for bread made with CB. Thus, it would appear that differences in fibre content between fractions is not the major determinant of differences in bread quality observed in our study. Cai et al. [4] found no differences in total dietary fibre between coarse, medium and fine bran fractions in a hard white and hard red wheat but dough water absorption increased with finer bran particle size, in contrast to the opposite effect we observed. The increase in protein content of the finer fractions (Table 1) might be expected to improve loaf volume given its correlation with protein level [29] but in fact LV was worse in the finer bran breads. Phenolic compounds, in particular ferulic acid (FA) have been shown to have negative effects on gluten properties [4,7], however our study found increasing levels of FA did not lead to reduced volumes except for 425 which had the highest FA content (Figure 3). Grinding the CB into a finer particle size could increase the availability of FA by releasing it from the bran matrix [30]. The resultant increased reactivity of the FA in finer fractions could therefore help explain the lower loaf volumes attained despite lower FA content.

Instrumental assessment (colour parameters, firmness, chewiness, resilience and cohesiveness) of bread enriched with bran fractions shows detrimental effects on bread properties. Fine particle size (180, <180) had the least effect on crumb brightness and redness compared with other bran fractions, also reported by Kim et al. [8]. This could be related to these fractions having more starch (endosperm) and less lignin. Moreover, finer fractions (180, <180) have low levels of phenolic acids compared with the coarser fractions like 425 and our study found a strong trend with increasing phenolic content of the fractions and increased crumb redness (*a**) and decreased brightness (*L**) (r = 0.87, −0.89, *p* < 0.05, respectively). Phenolic acids are known to be easily oxidised (polyphenol oxidase, metal, heat or high pH catalyse this oxidation) resulting in formation of quinones, compounds which react with amino acids and protein through Maillard reaction causing darkness of final product [31]. This enzymic browning arising from the ready oxidation of phenolic acids [32] resulted in a duller, redder appearance of the crumb. Crumb yellowness (*b**) did not exhibit identifiable trends across fractions but CB was significantly less yellow than all other fractions.

Fine fractions (180 and <180) made bread that was firmer and with higher chewiness than coarse bran fractions (Table 3). The finer fractions produced a denser crumb with more compact gas cell structure leading to a lower specific volume and firmer crumb. Other researchers reported increased crumb firmness using fine bran [4,33]. Bread enriched with coarse bran showed less effect on springiness, resilience and cohesiveness compared with fine fractions which decreased these properties significantly. In a recent study wholemeal bran obtained from grinding wholegrain wheat into coarse, medium and fine fractions with mean size of 1315, 450, and 199 µm was added to a bread mix. The coarse bran bread had a more compact structure, smaller specific volume and harder texture than bread made from the fine fraction [34] which is opposite to results in Table 3 and Table 4. The range of bread formulations and baking methodologies used in various studies may help explain the divergence in findings. Although lipid content was not measured, it is known that bran is rich in non-polar lipids, which have a detrimental effect on loaf volume [35]. If the concentration of non-polar lipids varied significantly between fractions, this may have contributed to reduced volume and increased crumb firmness.

## 5. Conclusions

Reduction in median bran particle size changed composition where the finer fractions (180 and <180) had the highest protein and starch contents compared to coarser bran fractions. The 180 and FB had the highest phytosterol contents, while CB and 425 were highest in AO with marginally higher TPA. We hypothesised that reduction in the particle size of wheat bran and inclusion in a bread mix has more negative impacts on bread quality than a broad range particle size bran fraction. Our results showed that grinding CB then sieving to create smaller median particle size fractions more negatively impacted bread volume and texture compared to CB. Although bread volume and texture are important, they cannot completely predict consumer preference that could be worse after adding coarse bran into bread compared with bran of finer particle size. To achieve the best bread quality and phytochemical content, supplementation using coarse bran would be preferable to using finer bran fractions and some consumers might prefer such bread if marketed as having health benefits.

## Figures and Tables

**Figure 1 foods-10-00489-f001:**
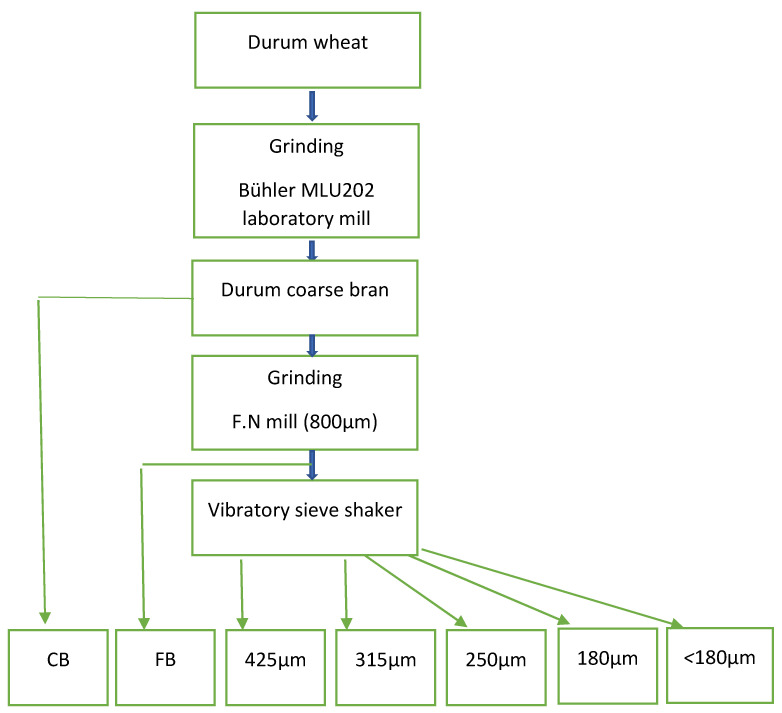
Flow chart for preparing durum bran fractions. F. N = falling number mill. CB; coarse bran. FB; fine bran.

**Figure 2 foods-10-00489-f002:**
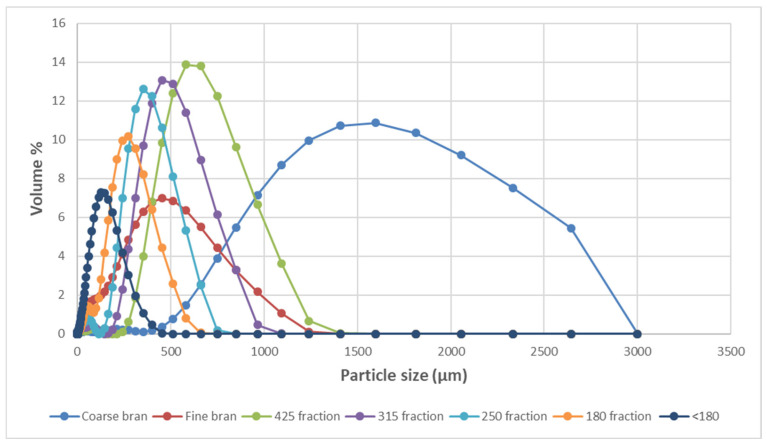
Particle size distribution of bran fractions coarse bran, fine bran, 425, 315, 250, 180, and <180.

**Figure 3 foods-10-00489-f003:**
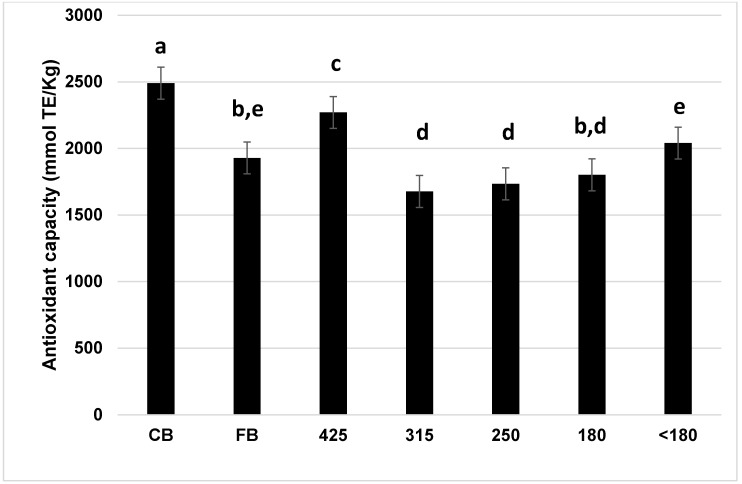
Antioxidant capacity (mmol TE/kg) of bran fractions. Bars labelled with the same letters are not significantly different, *p <* 0.001. CB; coarse bran. FB; fine bran.

**Figure 4 foods-10-00489-f004:**
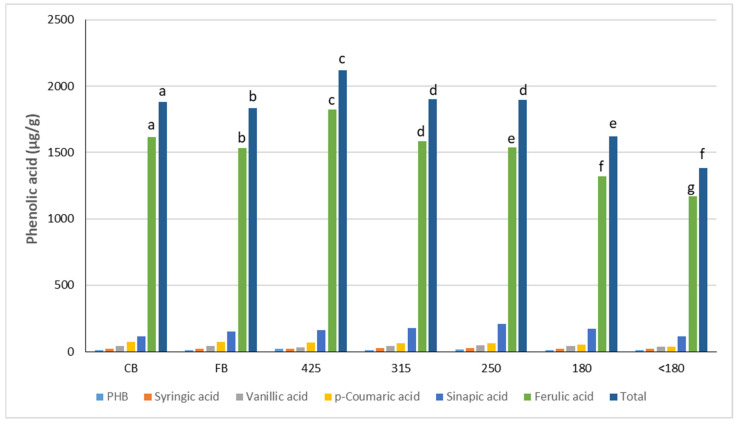
Phenolic acid content (μg/g dm) of bran fractions. Bars with alike letters in matching colours are not significantly different, *p* < 0.001. PHB = *p*-hydroxybenzoic acid. CB; coarse bran. FB; fine bran.

**Figure 5 foods-10-00489-f005:**
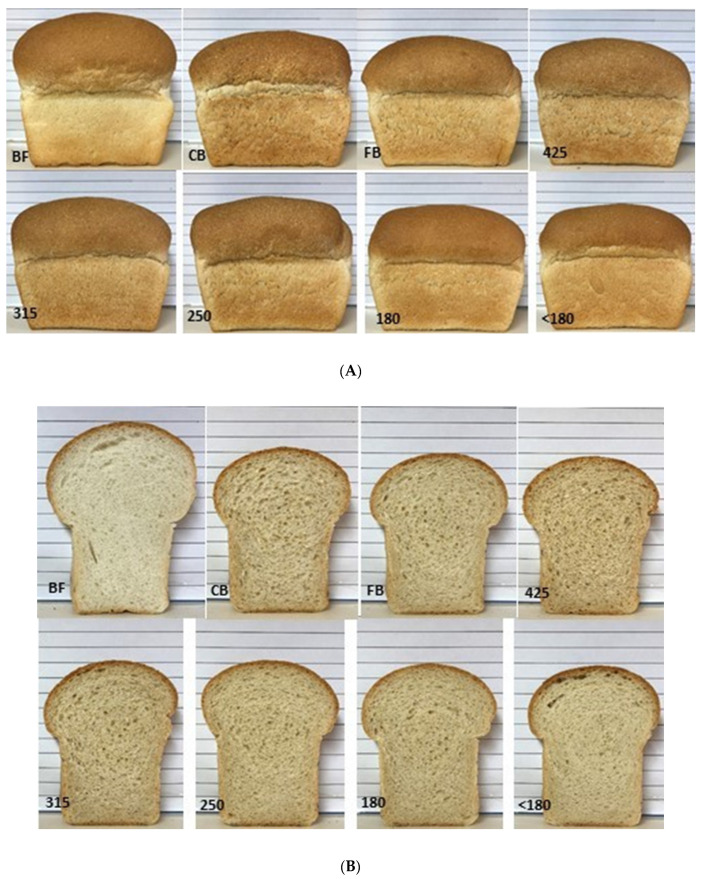
Loaves prepared with 100% Bakers flour (BF) and with 10% (*w*/*w*) bran fractions coarse bran (CB), fine bran (FB), 425, 315, 250, 180, and <180 µm. (**A**): External view; (**B**): Cross-sectional view.

**Table 1 foods-10-00489-t001:** Chemical composition of bran fractions (g/100 db). Data are mean ± stdev.

Sample	D50 (µm)	Moisture	* Protein	Ash	Total Starch
CB	1497	13.1 ± 0.32	15.6	3.2 ± 0.04	9.0 ± 0.17
FB	350	10.6 ± 0.14	17.2	3.1 ± 0.03	9.4 ± 0.04
425	641	10.8 ± 0.16	11.9	4.2 ± 0.02	7.3 ± 0.74
315	481	10.8 ± 0.21	13.3	3.6 ± 0.01	15.9 ± 0.42
250	357	11.2 ± 0.15	15.2	3.5 ± 0.18	26.0 ± 1.35
180	247	11.2 ± 0.21	16.6	4.1 ± 0.77	28.2 ± 0.66
<180	115	11.2 ± 0.10	19.2	3.9 ± 0.01	29.7 ± 1.47

* Measurement done in commercial lab with precision of measurement (cv) for protein of 1.3%. CB; coarse bran. FB; fine bran.

**Table 2 foods-10-00489-t002:** Phytosterol content (mg/g db) of bran fractions.

Sample	D50 (µm)	Campesterol	Campestanol	Stigmasterol	β-Sitosterol	Sitostanol	∆5-Avenasterol	Total
CB	1497	0.204 ^a^	0.187 ^a^	0.027 ^a^	0.502 ^a^	0.185 ^a^	0.079 ^a^	1.184 ^ac^
FB	350	0.337 ^b^	0.323 ^b^	0.048 ^a^	0.913 ^b^	0.310 ^b^	0.125 ^b^	2.055 ^b^
425	641	0.164 ^a^	0.287 ^b^	0.042 ^a^	0.414 ^a^	0.202 ^a^	0.055 ^c^	1.164 ^ac^
315	481	0.167 ^a^	0.201 ^a^	0.034 ^a^	0.403 ^a^	0.152 ^ac^	0.056 ^cd^	1.014 ^ac^
250	357	0.210 ^a^	0.219 ^a^	0.036 ^a^	0.558 ^a^	0.168 ^ac^	0.073 ^ad^	1.263 ^a^
180	247	0.342 ^b^	0.318 ^b^	0.119 ^b^	0.927 ^b^	0.272 ^b^	0.127 ^b^	2.107 ^b^
<180	115	0.153 ^a^	0.147 ^c^	0.046 ^a^	0.431 ^a^	0.128 ^c^	0.055 ^c^	0.958 ^c^

Numbers with alike superscript letters in same column are not significantly different, *p* < 0.001. CB; coarse bran. FB; fine bran.

**Table 3 foods-10-00489-t003:** Loaf volume, specific volume and oven spring of Baker’s flour bread enriched with bran fractions at 10% *w/w.*

Samples	Loaf Volume (cm^3^)	Specific Volume (cm^3^/g)	% Oven Spring
BF (control)	752 ^a^	4.89 ^a^	20.20 ^a^
CB	666 ^b^	4.30 ^b^	11.70 ^b^
FB	629 ^ce^	4.03 ^c^	8.45 ^c^
425	604 ^d^	3.83 ^df^	6.20 ^cd^
315	635 ^ce^	4.06 ^c^	7.40 ^c^
250	642 ^c^	4.10 ^c^	8.15 ^c^
180	621 ^e^	3.91 ^d^	4.05 ^de^
<180	598 ^d^	3.76 ^f^	3.65 ^e^

Numbers with alike superscript letters in same column are not significantly different, *p* < 0.001. BF; Baker’s flour. CB; coarse bran. FB; fine bran.

**Table 4 foods-10-00489-t004:** Bread crumb colour and texture characteristics of loaves prepared from control (100% Baker’s flour, BF) and 10% incorporated bran fractions.

Samples	Colour			Texture				
	*L**	*a**	*b**	Firmness	Springiness	Resilience	Cohesiveness	Chewiness
BF	78.9 ^a^	−0.1 ^a^	14.7 ^a^	586 ^a^	0.94 ^a^	0.34 ^a^	0.69 ^a^	381.0 ^a^
CB	73.2 ^b^	2.05 ^b^	16.5 ^b^	750 ^bd^	0.93 ^a^	0.31 ^b^	0.67 ^b^	463.3 ^b^
FB	73.7 ^c^	1.95 ^be^	17.7 ^c^	924 ^cd^	0.92 ^ab^	0.28 ^c^	0.64 ^c^	538.3 ^bc^
425	70.8 ^d^	3.05 ^c^	17.9 ^cd^	931 ^cd^	0.89 ^b^	0.28 ^c^	0.65 ^d^	538.3 ^bc^
315	71.9 ^e^	2.55 ^d^	17.7 ^c^	874 ^d^	0.91 ^b^	0.29 ^c^	0.65 ^d^	519.0 ^bc^
250	73.4 ^b^	1.95 ^b^	17.8 ^c^	905 ^cd^	0.92 ^ba^	0.28 ^c^	0.64 ^c^	534.5 ^bc^
180	74.0 ^f^	1.8 ^e^	17.9 ^cd^	1055 ^c^	0.91 ^b^	0.26 ^d^	0.63 ^e^	594.5 ^c^
<180	75.1 ^g^	1.45 ^f^	18.1 ^d^	1272 ^e^	0.89 ^b^	0.26 ^d^	0.62 ^f^	706.3 ^d^

Numbers with alike superscript letters in same column are not significantly different, *p <* 0.001. CB; coarse bran. FB; fine bran.

## Data Availability

The data presented in this study are available on request from the corresponding author. The data are not publicly available due to institutional privacy requirements.

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
