# Peer review of "Influence of Durum Wheat Bran Particle Size on Phytochemical Content and on Leavened Bread Baking Quality"

_foods, 2021, doi:10.3390/foods10030489_

Round 1
Reviewer 1 Report
The study 'Influence of durum wheat bran particle size on phytochemical content and on leavened bread baking quality' provides an insight in using durum wheat bran in making bread depending on its particle size and composition. The manuscript is generally well written but some information on methods are missing. Some results are missing and some are shown without measuring units. Also, the authors should try to relate the studied phenolic acids and phytosterols of bran with bread quality parameters. For valid conclusions on which bran particle size is optimal for bread making, sensory study is missing. Further, it is not clear to what extent would be advantageous to use durum wheat bran in bread making from nutrition perspective. More detail comments are given bellow.
Introduction:Please consider including the study by Sanz‐Penella, J. M., Laparra, J. M., Sanz, Y., & Haros, M. (2012). Influence of added enzymes and bran particle size on bread quality and iron availability. Cereal chemistry, 89(5), 223-229.
Lines 122-123: Please justify why T. aestivum flour was used and not durum wheat flour, since bran was from durum wheat.
Line 132-133: Was the water addition adjusted for each sample depending on the used bran?
Line 146: I do not see results for crust colour.
Line 148: Please provide parameters used in texture profile analysis or provide a reference for the method. How many replicates of texture and colour analyses were performed?
Table 1. The content of lipids and/or dietary fiber in bran fractions would be useful for interpretation of their effect on bread quality.
Table 4. The measuring units are missing for several texture properties.
Fig 3. Please increase the font on the axes and make the graph smaller.
Lines 339-340: Please consider the effect of lipids, sterols and phenolics on dough rheology and bread quality.
Conclusion: Although bread volume and texture are important bread quality features, they cannot completely predict the consumers preference, e.g. mouthfeel could be seriously worsen after adding coarse bran into bread compared with bran of finer particle size.
Author Response
We provide our response to each point by the reviewer. Line numbers Lxx refer to position in the revised manuscript.
Introduction:Please consider including the study by Sanz‐Penella, J. M., Laparra, J. M., Sanz, Y., & Haros, M. (2012). Influence of added enzymes and bran particle size on bread quality and iron availability. Cereal chemistry, 89(5), 223-229.
We have cited this reference at L398
Lines 122-123: Please justify why T. aestivum flour was used and not durum wheat flour, since bran was from durum wheat.
Although durum flour can be used for bread production, it does introduce its own quality defects including loaf volume reduction and increased crumb firmness. Therefore to clearly delineate the effect of bran fraction addition on bread quality, high quality bakers flour was used in this study. It should also be noted that bread made from common wheat flour is much more popular than bread made from durum wheat. Therefore, the application of bran in common wheat flour bread baking has more applications. We chose durum bran rather than common wheat bran because we wanted to see if we could find alternative uses of this low value by-product of durum milling and that no studies have reported on our approach. We have added to the text at L 229-230 "to clearly delineate the effect of bran addition on bread quality, high quality bakers flour was used in this study. "
Line 132-133: Was the water addition adjusted for each sample depending on the used bran?
Yes. Bran addition and its particle size distribution affects flour water absorption which was measured and used to calculate bake water absorption and the amount of water to add. This information has been included L148 "Bake water absorption for each bran fraction blend was calculated from farinograph water absorption corrected to 14% moisture basis and was FWA minus three."
Line 146: I do not see results for crust colour.
This is an error as only bread crumb colour was measured (table 4) so we have removed the term crust colour from the document. we do not think it would add anything to the paper to include crust colour results/discussion
Line 148: Please provide parameters used in texture profile analysis or provide a reference for the method. How many replicates of texture and colour analyses were performed?
We have provided more detail L167-175. "UK), fitted with a 35mm flat cylindrical probe and test speed setting of 1mm/sec, hold time 3 secs, and compression distance of 40%. Parameters measured were firmness (g), chewiness (undefined), resilience (%) and cohesiveness (%) according to Texture Technologies Corp and Stable Micro Systems standard TPA method Chapter IV (2018) https://texturetechnologies.com/resources/texture-profile-analysis. Moisture was determined using two-stage bread moisture method AACC Approved Method 44-15.02 (18) and crumb colour was measured using Minolta CR-410 with 50 mm head. Baking was performed in triplicate on a single day, providing six loaves of each sample for full analysis including comparison BF loaves. " There is no reference. We have also updated T4.
Table 1. The content of lipids and/or dietary fiber in bran fractions would be useful for interpretation of their effect on bread quality.
We did not measure these. DF was too expensive as it would need to be done by a contract laboratory and study funds for the student project did not allow this. Also, Cai et al. (4) found no differences in total dietary fibre between coarse, medium and fine bran fractions in a hard white and hard red wheat to justify. See comments about lipids at end of discussion.
Table 4. The measuring units are missing for several texture properties.
see above comment, now included
Fig 3. Please increase the font on the axes and make the graph smaller.
We have edited Fig 3
Lines 339-340: Please consider the effect of lipids, sterols and phenolics on dough rheology and bread quality.
We have already covered phenolics, specifically ferulic acid in the discussion L371- and 387-. For lipids, we have added information L410-414 and included a reference. We could not find any references relating sterols to bread quality which we feel is outside the scope of this study.
Conclusion: Although bread volume and texture are important bread quality features, they cannot completely predict the consumers preference, e.g. mouthfeel could be seriously worsen after adding coarse bran into bread compared with bran of finer particle size.
We have modified the conclusion L412-424

Reviewer 2 Report
December 31, 2020
Manuscript ID: foods-1064581
Type of manuscript: Article
Title: Influence of durum wheat bran particle size on phytochemical content and leavened bread baking quality
Opinion
The objective of the study was to characterize the physicochemical properties and phytochemical composition of durum bran fractions, and show how these and the particle size impact leavened bread technological properties through their inclusion in a common wheat bread mix.
Though the study included important laboratory techniques, the overall content of the manuscript is too wordy. The abstract is quite confusing and not providing a suitable reason why the study was planned? The Discussion part is too lengthy, and the conclusions sections contain unnecessary information. The manuscript in this form is not suitable to understand correctly however, I read and have some comments:
Comments:
Comment#1: Lines 15-16 are difficult to understand, if the bran is used in the feed industry (as mentioned in Line no.29), why the authors connected it with the bread making. The authors should justify both the sentences.
Comment#2: Please write the purpose of the study in the abstract.
Comment#3: The result section of the abstract is not to-the-point, it must be straight and the conclusions must be summarized in one to two lines in brief. (Line no. 22-24)
Comment#4: I suggest the authors to mention the full forms of all the units when they arise the first time in the text. For example Line no. 29; mmt
Comment#5: The discussion part again too much extended. No need to summarize cited studies, when the authors have enough material from the study to mention. The authors should stick to explaining their results. In my view, several lines can be trimmed. In this form, it gives a feel of review article.
Comment#6: In this line, the authors have mentioned ‘there are multiple studies’; however, they cited only one study for this. They should cite more than one study to justify the line. (Line no. 301)
Comment#7: The Conclusion part is unnecessary stretched. It must be straight-to-the-point, only the main findings of the study should be reflected. Please remove the statistical information.
Other comments
Comment#8: Add a reference for this line (……developing cardiovascular disease and cancer). (Line no. 263)
Comment#9: There are several grammatical and punctuation mistakes in the manuscript. I hope the authors would identify and correct them.
Author Response
Though the study included important laboratory techniques, the overall content of the manuscript is too wordy. The abstract is quite confusing and not providing a suitable reason why the study was planned? The Discussion part is too lengthy, and the conclusions sections contain unnecessary information. The manuscript in this form is not suitable to understand correctly however, I read and have some comments:
We have revised the abstract make clear the objectives. The discussion has been reduced but we feel to explain the results, a detailed proposed explanation is needed supported with relevant literature and this should not be deleted.
Comments:
Comment#1: Lines 15-16 are difficult to understand, if the bran is used in the feed industry (as mentioned in Line no.29), why the authors connected it with the bread making. The authors should justify both the sentences.
We have re-written the abstract L15-28 Wheat bran is a conventional by-product of the wheat milling industry mainly used for animal feed. It is a rich and inexpensive source of phytonutrients so is in demand for fibre rich food products but creates quality issues when incorporated into bread. The purpose of this study was to characterize the physicochemical properties and phytochemical composition of different size durum bran fractions, and show how they impact bread quality. Durum wheat (Triticum durum Desf.) was milled to create a coarse bran fraction (CB) which was further ground into a finer fraction (FB) which was sieved using four screens with apertures 425, 315, 250, 180 and <180 µm to create a particle size range of 1497 to 115 µm. All fractions contained phytosterol with highest in the 180 and FB, while total phenolic and antioxidant capacity was highest in CB and 425. Use of the fractions in a leavened common wheat (T. aestivum L.) bread formula at 10% incorporation negatively impacted bread loaf volume, colour and texture compared to standard loaves, with CB having the least impact. Results suggest that to combine highest phytochemical content with minimal impact on bread quality, bran particle size should be considered with CB being the best choice.
Comment#2: Please write the purpose of the study in the abstract.
We have included this at L17-19 "The purpose of this study was to characterize the physicochemical properties and phytochemical composition of different size durum bran fractions, and show how they impact bread quality. "
Comment#3: The result section of the abstract is not to-the-point, it must be straight and the conclusions must be summarized in one to two lines in brief. (Line no. 22-24)
We have re-written the abstract L15-28 as above
Comment#4: I suggest the authors to mention the full forms of all the units when they arise the first time in the text. For example Line no. 29; mmt
We have done this throughout the document for unfamiliar unit abbreviations. It is not necessary for common ones like mm, mL, min etc.
Comment#5: The discussion part again too much extended. No need to summarize cited studies, when the authors have enough material from the study to mention. The authors should stick to explaining their results. In my view, several lines can be trimmed. In this form, it gives a feel of review article.
The discussion has been reduced but we feel to explain the results, a detailed proposed explanation is needed supported with relevant literature. We have removed details on cited studies.
Comment#6: In this line, the authors have mentioned ‘there are multiple studies’; however, they cited only one study for this. They should cite more than one study to justify the line. (Line no. 301)
All studies are listed in the review so we have re-written to "There are multiple studies evaluating the impact of fibres on dough stability with conflicting results reviewed by Ktenioudaki and Gallagher (24)."
Comment#7: The Conclusion part is unnecessary stretched. It must be straight-to-the-point, only the main findings of the study should be reflected. Please remove the statistical information.
We have shortened this section
Other comments
Comment#8: Add a reference for this line (……developing cardiovascular disease and cancer). (Line no. 263)
we have included a reference for CVD and cancer-reference #2
Comment#9: There are several grammatical and punctuation mistakes in the manuscript. I hope the authors would identify and correct them.
we have edited the manuscript to improve this
Round 2
Reviewer 1 Report
The authors have improved their manuscript to satisfactory level. I have only one remark: the authors have used the same abbreviation (TPA) for different terms - total phenolic acids (line 91) and texture profile analysis (line 127), which should be corrected.
Author Response
We thank the reviewer for their careful checking of the revision. We have changed L127 from TPA to texture profile analysis.
We also found use of term TPC which is meant to be TPA so have corrected at L252 and 360.
Mike Sissons
on behalf of my co-authors

Reviewer 2 Report
I have no further comments
Author Response
We thank the reviewer for their approval of the revisions. We have checked for spelling errors.
Edits made:
L23 phenolic acids
L27 insert ","
L93 extracts not extacts
L122 spacing
L132 inserted "100% wheat flour (BF) "
L184 insert "(BF)"
L250 inserted "elevated"
L341 inserted "made bread"
Mike Sissons
On behalf of my co-authors
